# PPIscreenML is a method for structure-based screening of protein-protein interactions using AlphaFold

**Victoria Mischley[1,2], Johannes Maier[3], Jesse Chen[3], John Karanicolas[1,4]\***

[1]Cancer Signaling and Microenvironment Program, Fox Chase Cancer Center, Philadelphia, United States; [2]Molecular Cell Biology and Genetics, Drexel University, Philadelphia, United States; [3]Triana Biomedicines, Lexington, United States; [4]Moulder Center for Drug Discovery Research, Temple University School of Pharmacy, Philadelphia, United States

## eLife assessment

This study explores simple machine learning frameworks to distinguish between interacting and non-interacting protein pairs, offering **solid** computational results despite some concerns about dataset generation. The authors demonstrate a modest improvement in AlphaFold-multimers' ability to differentiate these pairs. Using a simple yet sound approach, this work is a **valuable** contribution to the challenging problem of reconstructing protein-protein interaction networks.

**\*For correspondence:** johnkaranicolas1@gmail.com

**Sent for Review** 08 April 2024

**Preprint posted** 30 April 2024

**Reviewed preprint posted** 08 July 2024

**Version of Record published** 21 April 2026

**Abstract** Protein-protein interactions underlie nearly all cellular processes. With the advent of protein structure prediction methods such as AlphaFold2 (AF2), models of specific protein pairs can be built extremely accurately in most cases. However, determining the relevance of a given protein pair remains an open question. It is presently unclear how to use best structure-based tools to infer whether a pair of candidate proteins indeed interacts with one another: ideally, one might even use such information to screen among candidate pairings to build up protein interaction networks. Whereas methods for evaluating quality of modeled protein complexes have been co-opted for determining which pairings interact (e.g. pDockQ and iPTM), there have been no rigorously bench-marked methods for this task. Here, we introduce PPIscreenML, a classification model trained to distinguish AF2 models of interacting protein pairs from AF2 models of compelling decoy pairings. We find that PPIscreenML outperforms methods such as pDockQ and iPTM for this task, and further that PPIscreenML exhibits impressive performance when identifying which ligand/receptor pairings engage one another across the structurally conserved tumor necrosis factor superfamily (TNFSF). Analysis of benchmark results using complexes not seen in PPIscreenML development strongly suggests that the model generalizes beyond training data, making it broadly applicable for identifying new protein complexes based on structural models built with AF2.

## Introduction

Cellular homeostasis is maintained by a complex protein interaction network that contributes to intricate signaling pathways within the cell (*Choi et al., 2019*). As the human interactome is estimated to contain between 74,000 and 200,000 protein interactions (*Venkatesan et al., 2009*), accurate and high-throughput methods to identify these interactions are crucial (*Elhabashy et al., 2022*). Classically, discovery and cataloging of protein-protein interactions (PPIs) have most commonly employed yeast two-hybrid assays (*Fields and Song, 1989*) and affinity purification assays (*Dunham et al.,*

2012). More recently, proximity labeling techniques such as BioID (*Roux et al., 2018*) and TurboID (*Cho et al., 2020*) have gained popularity. However, each method has its own specific shortcomings, which are further compounded due to sensitive dependence on experimental techniques and cellular conditions (*Choi et al., 2019*; *Qin et al., 2021*). Collectively, the variabilities in these approaches ultimately lead to high rates of both false positives and false negatives (*Qin et al., 2021*; *Deane et al., 2002*). For example, statistical analysis of representative interactome mapping data showed that recovering 65% of true protein interactions would require combining 10 distinct assays (*Choi et al., 2019*).

Computational predictions offer a natural means for complementing these experimental techniques. Early methods focused on inferring PPIs based on sequence information, by searching for gene pairs with correlated expression patterns, or distinct genes that can be found fused together in a different organism, or distinct genes that co-evolve with one another (*Marcotte et al., 1999*). Drawing upon the exciting successes of large language models in many contexts, deep learning methods have also been applied to infer binary PPI pairings using protein sequences (*Zhao et al., 2022*; *Jha et al., 2023*; *Du et al., 2017*). Results from such methods are extremely promising; however, their lack of explainability makes it difficult to determine how such models might perform on PPIs that are dissimilar from the training examples (i.e. their domain of applicability). Moreover, these models are inherently not designed to provide structural information about the bound pose: this information can be extremely valuable, both to provide biological understanding that spurs further experimentation and as a starting point for developing new therapeutic agents that modulate the PPI (*Lu et al., 2020*).

Broad availability of AlphaFold2 (AF2) (*Jumper et al., 2021*), especially through the ColabFold framework (*Mirdita et al., 2022*), has provided the research community with unprecedented access to structural information for monomeric proteins, in many cases at resolution comparable to experimentally determined structures. The startling accuracy of certain AF2 predictions, in turn, enabled structure-based modeling for numerous new applications in biology and medicine (*Akdel et al., 2022*; *Yang et al., 2023*). Whereas the original AF2 could be adapted for highly successful predictions of oligomeric assemblies, a subsequent retrained release dubbed AF2-Multimer provided further enhanced performance in this regime (*Evans et al., 2021*).

While AF2 provides often-accurate predictions for interacting protein pairs, there is not a direct measure to evaluate the likelihood that a given query pair indeed interacts with one another – a set of predicted output structures will be returned regardless. AF2 also provides residue-level confidence measures of the predicted structures, specifically the predicted local-distance difference test (pLDDT) (*Mariani et al., 2013*), the predicted TM-score (pTM) (*Zhang and Skolnick, 2005*), and the predicted aligned error (PAE). The accuracy of structures for predicted protein complexes is often defined using DockQ (*Basu and Wallner, 2016*), a metric that combines three complementary and frequently used quality measures. By combining the number of interface residues with the pLDDT values of interface residues, a confidence measure pDockQ was developed to define the likelihood of an accurate binding mode in protein complexes predicted by AF2 (*Bryant et al., 2022*). Interestingly, pDockQ also provided some ability to discriminate between 'true' (interacting) and 'decoy' (noninteracting) protein pairs (*Bryant et al., 2022*) – performance for this task was only modest, however, presumably because development of pDockQ was fundamentally focused elsewhere (recapitulating DockQ values).

In light of interest in using AF2 to distinguish between interacting and noninteracting protein pairs, the AlphaPulldown (*Yu et al., 2023*) Python package calculates several measures that have been proposed for this task: iPTM (*Evans et al., 2021*), PIscore (*Malhotra et al., 2021*), and pDockQ (*Bryant et al., 2022*). While each of these measures is intuitively reasonable, none were expressly developed for this classification task.

Here, we describe PPIscreenML, a machine learning classifier that uses AF2 models to distinguish interacting versus noninteracting protein pairs. To do so, PPIscreenML uses descriptors drawn from AF2 confidence measures (*Jumper et al., 2021*) and complements these with energetic terms from the Rosetta scoring function (*Leman et al., 2020*). In contrast to methods such as pDockQ, which was trained as a means of evaluating model quality, PPIscreenML is deliberately and intentionally trained/tested to distinguish between models of interacting proteins and compelling decoy pairs. PPIscreenML exhibits superior performance to other methods for identifying interacting pairs in a retrospective screen. Finally, using the tumor necrosis factor superfamily (TNFSF) as an example of a

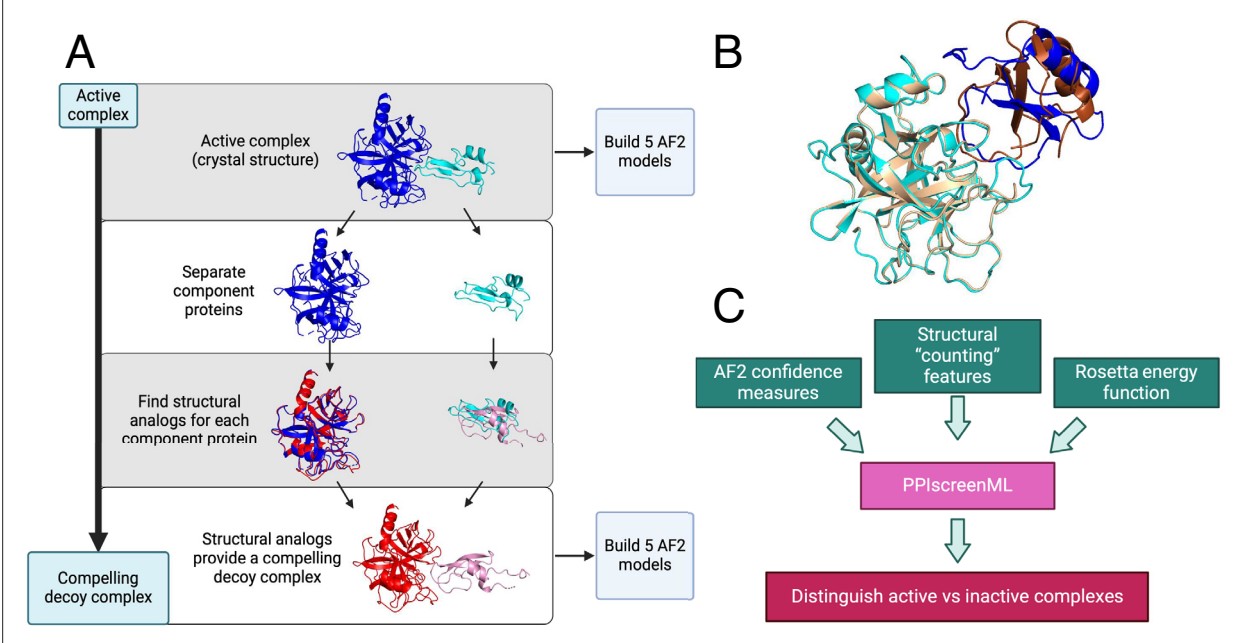

**Figure 1.** Building a challenging training/testing set for PPIscreenML. (**A**) A collection of 1481 nonredundant active complexes with experimentally derived structures was obtained from DockGround, and five AlphaFold2 (AF2) models were built from each of these. To build decoys, the same collection was screened to identify the closest structural matches (by TM-score) for each component protein. The structural homologs for each template protein were aligned onto the original complex, yielding a new (decoy) complex between two presumably noninteracting proteins. Five AF2 models were built for each of these 1481 decoy complexes. (**B**) An example of a decoy complex (*blue/cyan*) superposed with the active complex from which it was generated (*brown/wheat*). (**C**) A suite of AlphaFold confidence metrics, structural properties, and Rosetta energy terms were used as input features for training PPIscreenML, a machine learning classifier built to distinguish active versus compelling inactive protein pairs.

The online version of this article includes the following figure supplement(s) for figure 1:

**Figure supplement 1.** Composition of the active and decoy datasets.

**Figure supplement 2.** Overview of dataset construction and evaluation.

**Figure supplement 3.** List of 57 total features considered when training PPIscreenML.

**Figure supplement 4.** Overlaid histogram of total sequence length between actives and decoys.

structurally conserved family of ligand/receptor pairings, we demonstrate that PPIscreenML can accurately recapitulate the selectivity profile in this challenging regime.

## Computational approach

### Dataset for training/testing PPIscreenML

Critical to development of a useful classification model is compiling a set of 'active' and 'decoy' examples that accurately reflect those that will be encountered in future prospective applications. For PPIscreenML, we employed an overarching strategy that mirrors our previous studies leading to a machine learning classifier that identifies active protein/small molecule complexes likely to engage one another (*Adeshina et al., 2020*). Specifically, our approach seeks to build highly *compelling* decoy complexes, so that the resulting model is challenged to distinguish actives from decoys that cannot be resolved using solely trivial criteria.

To build relevant and diverse structures of active protein pairings (*Figure 1A*), we used the Dock-Ground database (*Collins et al., 2022*) to assemble the complete set of heterodimeric protein complexes in the PDB, excluding homodimers, antibody/antigen complexes, and complexes that were either very large or very small (see *Methods*). Upon filtering for a maximum of 30% sequence identity, there remained 1481 nonredundant protein-protein complexes with experimentally defined structures. The planned application of PPIscreenML is to distinguish AF2 models of active pairs from AF2 models of inactive pairs – thus, the model should be trained on AF2 actives, rather than

experimentally defined structures. Accordingly, we used the sequences of the component proteins to build five AF2 models for each of these 1481 nonredundant complexes.

Active protein-protein pairs that are mis-predicted by AF2 should not lead to predictions of 'active' from PPIscreenML, since the specific interactions in the AF2 are not those that lead to binding. In the context of training, then, we sought to use only those models that were correctly predicted by AF2: we therefore filtered the models and retained only those with a DockQ score (*Basu and Wallner, 2016*) of at least 0.23 relative to the corresponding experimentally defined structure (0.23 is the quality cutoff defined to be an 'acceptable' prediction for the MOAL and CAPRI sets). Filtering on model quality in this manner excluded 932 models from training, leading to a total of 6473 active models with median DockQ value of 0.88 (*Figure 1—figure supplement 1A*).

In a prospective application of PPIscreenML, however, the DockQ score will not be known a priori (since this requires an experimentally defined structure, which itself is evidence of an interaction). To ensure a realistic estimate of the performance that can be expected in future prospective experiments, complexes with structures that were mis-predicted by AF2 were *not* excluded from the test set. Accordingly, active pairs that are not correctly classified by PPIscreenML may occur either because AF2 builds a poor model, or because PPIscreenML fails to recognize a well-built model as an interacting pair. While the test set performance thus reflects more than the success of PPIscreenML, this strategy enhances confidence that the observed test set performance will carry forward to prospective applications in the future.

With respect to creating a set of decoy complexes (for both training and testing), it is essential that the decoys suitably represent the types of inactive complexes expected to be encountered in prospective applications. In particular, decoys that can be dismissed as inactive based on trivial criteria offer little benefit in developing PPIscreenML: rather, we prioritized building a set of extremely compelling decoys (*Adeshina et al., 2020*), so that training would be impelled to find a more sophisticated and nuanced model for classification.

To build a set of compelling decoys (*Figure 1A*), we began by using each active complex as a template. For each component protein in the complex, we identified its closest structural analog (based on TM-score; *Zhang and Skolnick, 2004*) among all other proteins in our collection. Despite the fact that homologs had been excluded from our dataset on the basis of sequence identity, close structural matches could nonetheless be identified in most cases (*Figure 1—figure supplement 1B*): 87% of the proteins in our set were paired with a structural analog having TM-score greater than 0.5, a cutoff far beyond that which is observed between random protein pairs (*Xu and Zhang, 2010*). The structural analog for each component protein was then superposed onto the template in the active complex, providing a crude model of these two (presumably) noninteracting proteins that structurally resembles an active complex (*Figure 1B*). As noted earlier, the planned application of PPIscreenML is to distinguish AF2 models of active pairs from AF2 models of inactive pairs: as for the active pairings, we therefore used the sequences of the component proteins in each of these 'decoy' complexes to build five AF2 models for each.

To train PPIscreenML, data was partitioned using a 60/20/20 split. The data reserved for testing was not used in any phase of training or validation. The dataset includes five AF2 models for each (active or decoy) complex: training on one AF2 model of a given complex then using a different model of the same complex in the test set would introduce obvious information leakage. To avoid this, all five models for a given complex were *together* placed in either the training set or the validation set or the test set (*Ong et al., 2023*). As noted earlier, all five AF2 models for a given complex were included in the validation and testing sets; however, only those AF2 models of active complexes close to the experimentally derived structures were included in the training set (*Figure 1—figure supplement 2*).

Having partitioned the AF2 models into training, validation, and test sets, a series of quantitative features were extracted from each model. Specifically, three groups of features were calculated from each model (*Figure 1C*): 7 features representing AF2 confidence measures, 33 structural 'counting' features, and 17 features from the Rosetta energy function. This set of 57 features (*Figure 1—figure supplement 3*) was initially evaluated in the development of PPIscreenML.

## Developing the PPIscreenML model

Using the compiled collection of features for each complex, we evaluated seven standard machine learning frameworks to build classification models. In each case, models were trained using fivefold

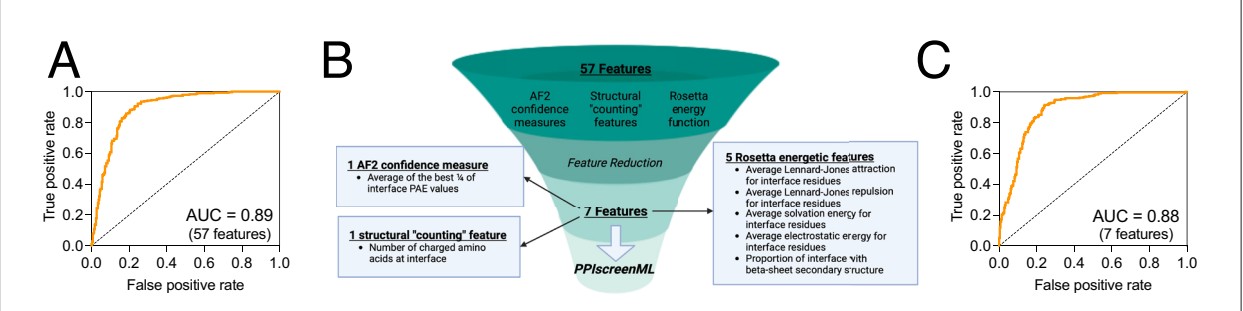

**Figure 2.** Training and feature reduction for PPIscreenML. (**A**) Receiver operating characteristic (ROC) plot demonstrating classification performance on a completely held-out test set, for an XGBoost model using 57 features. (**B**) The number of features was reduced using sequential backward selection, from 57 features to 7 features. Created with Biorender.com. (**C**) Classification performance of PPIscreenML (7 features) on the same completely held-out test set.

The online version of this article includes the following figure supplement(s) for figure 2:

**Figure supplement 1.** Comparisons of different machine learning classifiers.

**Figure supplement 2.** Feature reduction for PPIscreenML.

**Figure supplement 3.** PPIscreenML performance on models generated with AFPTM.

**Figure supplement 4.** PPIscreenML performance on models generated with AF-Multimer v2.2.

cross-validation within the test set and applied to a validation set that was not included in each fold of training. Models were then compared using the combined cross-fold AUC score (area under the curve of a receiver operating characteristic [ROC] plot). All seven models had similar performance, with XGBoost (an implementation of gradient-boosted decision trees) yielding the highest AUC value of 0.924 (*Figure 2—figure supplement 1*).

The model was then applied to the test set, which had been completely held out from model training and evaluation. To better recapitulate a real-world use case, this experiment incorporates two key differences from the earlier evaluation using the validation set (*Figure 1—figure supplement 2*). First, AF2 models of active complexes with poor DockQ scores relative to the corresponding PDB structure are included in the test set, whereas these had been excluded from the validation set. This reflects the reality that active complexes will occasionally be mis-docked by AF2, and these models cannot be discarded in a true prospective application. Second, rather than evaluating performance using the scores of individual AF2 models, the top-ranked model for each complex is drawn from the scores of all five AF2 models. Here too, in a prospective screening application, one would mitigate the possibility that individual AF2 models may be mis-docked by simply using the model that PPIscreenML finds most compelling. Applying the XGBoost model to the held-out test set with these conditions yielded an AUC value of 0.892 (*Figure 2A*).

To mitigate any potential overtraining, sequential backward selection was applied to reduce the number of features, using the validation set (*not* the test set, which is not used in any model optimization). This approach provided a model with only seven features that maintained the AUC of the model with all features when evaluated using the validation set (*Figure 2—figure supplement 2*). Specifically, this model uses one feature from the AF2 confidence measures (the average of the top ¼ of the interfacial PAE values), one structural 'counting' feature (number of charged amino acids in the interface), and five features from the Rosetta energy function (the average Lennard-Jones attractive score for interfacial residues, the average Lennard-Jones repulsive score for interfacial residues, the average solvation score for interfacial residues, the average electrostatic score for interfacial residues, and the proportion of interfacial residues involved in beta-sheet structures) (*Figure 2B*).

When applied to the test set that was completely held out from model training and optimization, this 7-feature PPIscreenML model yields an AUC of 0.884, comparable to the full 57-feature model (*Figure 2C*).

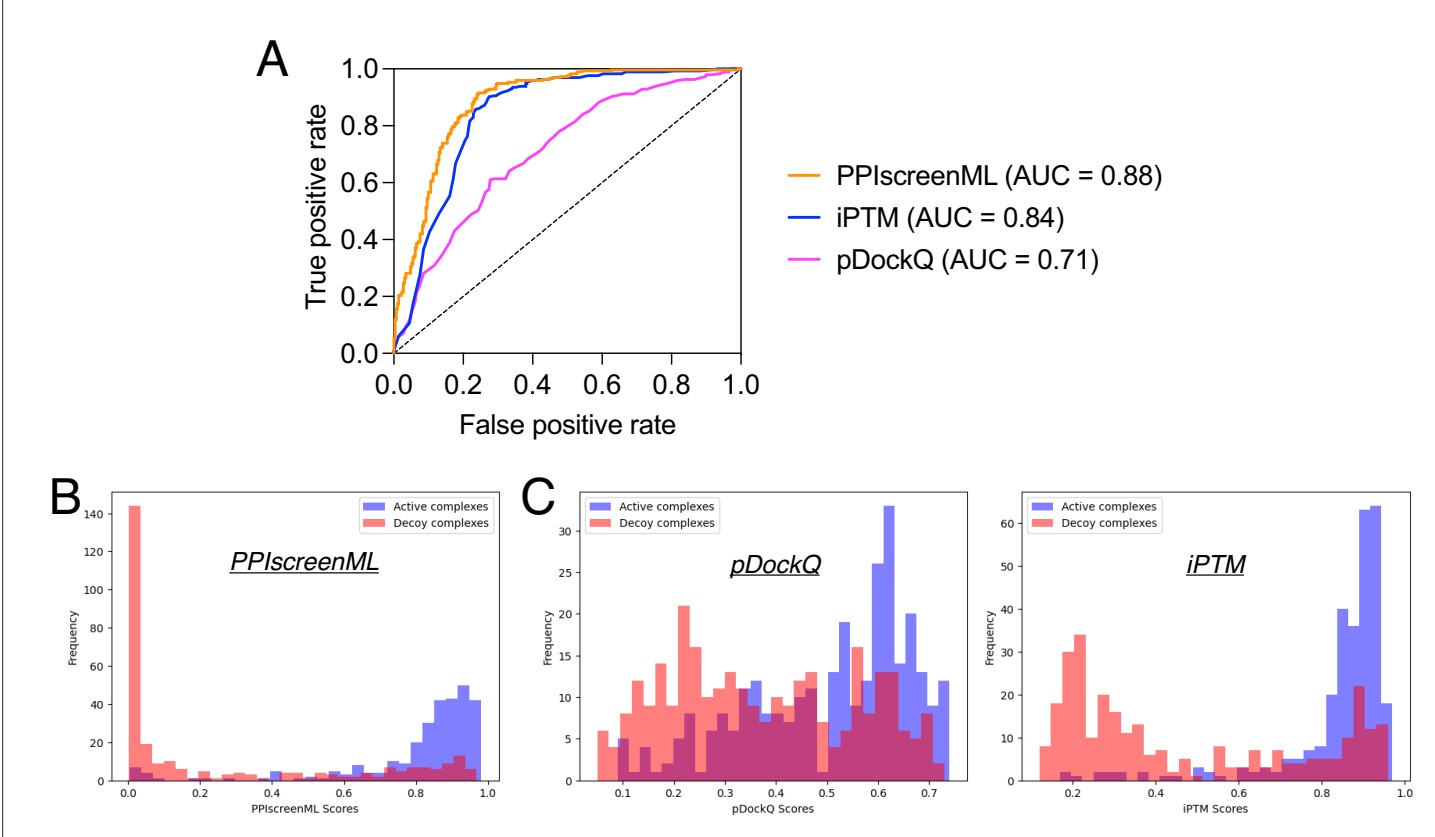

**Figure 3.** Classification performance of PPIscreenML relative to pDockQ and iPTM. The same test set is used here. These complexes were not seen in any phase of developing PPIscreenML, but may have been used in developing pDockQ or iPTM. (**A**) Receiver operating characteristic (ROC) plot shows superior performance of PPIscreenML relative to these other two methods. (**B**) Overlaid histograms show clear separation of actives and decoys scored using PPIscreenML. (**C**) Overlaid histograms show overlapping distributions when models are scored with pDockQ or iPTM.

## Results

### PPIscreenML outperforms other structure-based methods for identifying interacting protein pairs

Already several structure-based methods have been proposed as a means for distinguishing between models of interacting versus noninteracting protein pairs: these include iPTM (*Evans et al., 2021*), PIscore (*Malhotra et al., 2021*), and pDockQ (*Bryant et al., 2022*). To evaluate the performance of PPIscreenML relative to these tools, we applied each method to the collection of AF2 models for active pairings and compelling decoys that comprise our test set. As noted earlier, the test set does *not* remove models of true pairings for which the structures have been incorrectly built by AF2, since this would not be possible in a real-world setting. In each case, we scored all five AF2 models for a given pairing, and we took the highest score from each method as an indication of whether each pairing is inferred to be active or decoy (*Figure 1—figure supplement 2*). We note that none of the complexes in this test set have been used in any phase of PPIscreenML development, but we cannot rule out that some of the active complexes were used in training the other methods.

Applying each method to the same test set, we find that PPIscreenML outperforms the other methods: its AUC value is 0.884, compared to AUC values of 0.843 and 0.710 for iPTM and pDockQ, respectively (*Figure 3A*). The basis for these AUC values is evident from the distribution of raw scores assigned to active/decoy complexes by each method. PPIscreenML assigns (correct) scores very close to either 1 or 0 for most of the actives or decoys, with clear separation between them (*Figure 3B*); on the other hand, the alternate methods both assign more scores with intermediate values, and they also include many incorrect assignments (*Figure 3C*).

**Table 1.** Performance of PPIscreenML using various threshold values.

By adjusting the threshold score at which a test complex is assigned as active/decoy, PPIscreenML can be used in regimes that prioritize returning only the most confident pairings (a high threshold score yields high precision but poor recall) or in exploratory regimes that return more speculative pairings as well (a lower threshold score yields high recall but poorer precision).

| PPIscreenML threshold | FPR | TPR | FNR | TNR | Precision | Recall | F1 score |
|---|---|---|---|---|---|---|---|
| 0.98 | 0.00 | 0.00 | 1.00 | 1.00 | 1.00 | 0.00 | 0.01 |
| 0.95 | 0.01 | 0.16 | 0.84 | 0.99 | 0.96 | 0.16 | 0.27 |
| 0.9 | 0.07 | 0.39 | 0.61 | 0.93 | 0.85 | 0.39 | 0.54 |
| 0.85 | 0.11 | 0.60 | 0.40 | 0.89 | 0.85 | 0.60 | 0.70 |
| 0.8 | 0.15 | 0.74 | 0.26 | 0.85 | 0.83 | 0.74 | 0.78 |
| 0.75 | 0.18 | 0.80 | 0.20 | 0.82 | 0.82 | 0.80 | 0.81 |
| 0.7 | 0.21 | 0.85 | 0.15 | 0.79 | 0.80 | 0.85 | 0.83 |
| 0.64 | 0.23 | 0.88 | 0.12 | 0.77 | 0.80 | 0.88 | 0.84 |
| 0.6 | 0.24 | 0.89 | 0.11 | 0.76 | 0.79 | 0.89 | 0.84 |
| 0.56 | 0.24 | 0.92 | 0.08 | 0.76 | 0.79 | 0.92 | 0.85 |
| 0.5 | 0.27 | 0.93 | 0.07 | 0.73 | 0.78 | 0.93 | 0.85 |
| 0.47 | 0.27 | 0.93 | 0.07 | 0.73 | 0.77 | 0.93 | 0.84 |
| 0.4 | 0.29 | 0.95 | 0.05 | 0.71 | 0.76 | 0.95 | 0.85 |
| 0.38 | 0.30 | 0.95 | 0.05 | 0.70 | 0.76 | 0.95 | 0.85 |
| 0.28 | 0.33 | 0.95 | 0.05 | 0.67 | 0.75 | 0.95 | 0.84 |
| 0.21 | 0.35 | 0.95 | 0.05 | 0.65 | 0.73 | 0.95 | 0.83 |
| 0.18 | 0.35 | 0.96 | 0.04 | 0.65 | 0.73 | 0.96 | 0.83 |
| 0.08 | 0.43 | 0.96 | 0.04 | 0.57 | 0.69 | 0.96 | 0.81 |
| 0.05 | 0.48 | 0.97 | 0.03 | 0.52 | 0.67 | 0.97 | 0.79 |
| 0 | 1.00 | 1.00 | 0.00 | 0.00 | 0.50 | 1.00 | 0.67 |

An advantage of using ROC plots for defining the performance of binary classifiers is that one need not define a threshold score at which a given input would be assigned the label of active or inactive; rather, all points in the test set are simply ranked, and the ROC plot reflects the preponderance of actives that score ahead of decoys. To facilitate adoption of PPIscreenML in different settings, we use the test set results to calculate various performance measures (e.g. false positive rate, false negative rate) as a function of threshold score used to assign the label (*Table 1*). The use of a high (stringent) threshold value yields high precision but poor recall: this would be preferable in a very broad screening context in which one seeks (for example) to identify new candidate interactions at proteome-wide scale, and one wishes to test only the more confidently predicted pairings. Conversely, a lower (less stringent) threshold value yields poorer precision but improved recall: this would be preferable when screening in a more restricted space such as (for example) when seeking to identify additional pairings for members of a protein family known to interact with one another.

## PPIscreenML recapitulates selectivity within a protein superfamily

As a challenging further benchmark for PPIscreenML performance, we sought to evaluate whether it could recapitulate the pattern of binding interactions across a structurally conserved protein super-family. The TNFSF comprises 18 protein ligands and 28 receptors in humans, leading to 504 potential pairings. However, only 36 of these pairings are reported to show productive binding to one another, as observed in a comprehensive flow cytometry experiment (*Bossen et al., 2006*).

The TNFSF ligands share a conserved fold, and they assemble into trimers. The TNFSF receptors also share a conserved fold, and three copies of the cell-surface receptor independently bind around the

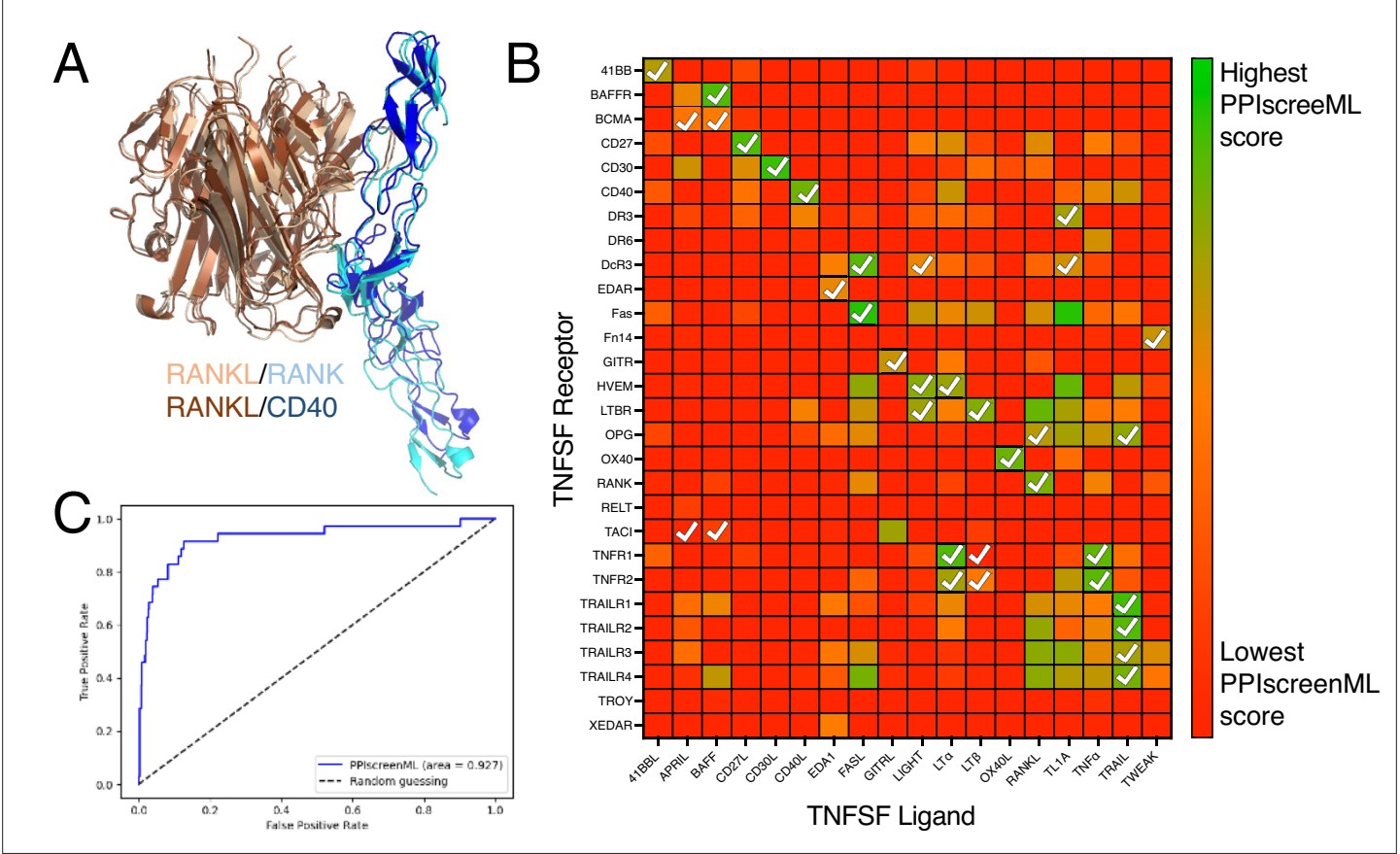

**Figure 4.** Application of PPIscreenML to identify active pairings within the tumor necrosis factor superfamily (TNFSF). (**A**) Structurally conserved TNFSF ligands bind to structurally conserved TNFSF receptors; AlphaFold2 (AF2) builds models of these complexes in the canonical pose for cognate pairings (RANKL/RANK are shown in *wheat/cyan*) but also in some cases for non-cognate pairings (RANKL/CD40 are shown in *brown/blue*). (**B**) Each ligand/receptor pairing was built with AF2 and scored with PPIscreenML (heatmap colored from low score in *red*, to high score in *green*). Ligand/receptor pairings observed in a comprehensive cellular assay are indicated with *white* checkmarks. (**C**) Receiver operating characteristic (ROC) plot demonstrating PPIscreenML classification of TNFSF ligand/receptor pairings.

edges of the trimeric ligand. While AF2 typically provides highly accurate structures of cognate ligand/receptor complexes, it also builds analogous models for ligand/receptor pairings that do not actually interact (*Figure 4A*). Accordingly, this system represents an ideal benchmark to evaluate whether PPIscreenML can use structural cues to distinguish active pairings from close analogs that do not bind one another. Importantly for this benchmark, no TNFSF ligands or receptors were included in the development of PPIscreenML thus far, because the training and testing sets were restricted to dimeric complexes.

We used AF2 to build models of all 504 potential TNFSF ligand/receptor pairings, and then evaluated these models using PPIscreenML. Overall, the ligand/receptor pairings that scored most favorably with PPIscreenML indeed correspond to those which have been described experimentally (*Figure 4B*). Of note, for 14 of the 18 TNFSF ligands, the receptor with the highest PPIscreenML score is indeed a true interactor; of the two receptors with the highest PPIscreenML score, a true interactor is found for 17 of the 18 ligands.

Analysis of these data via ROC plot confirms the outstanding performance of PPIscreenML for identifying interacting pairs (*Figure 4C*). Further, the AUC value of 0.93 in this new benchmark is similar to that observed when applying PPIscreenML to the held-out test set of heterodimeric complexes (0.88), supporting the generality of PPIscreenML for application to new data beyond the training regime.

## Discussion

The availability of AF2 and related methods (*Jumper et al., 2021*; *Mirdita et al., 2022*; *Evans et al., 2021*; *Baek et al., 2021*) has enabled structure-based analysis of proteins and protein complexes to

an extent that has rapidly transformed many parts of biology and medicine (*Akdel et al., 2022*; *Yang et al., 2023*). Numerous studies have incorporated AF2 models to justify or rationalize the structural basis for a particular PPI, and in many cases, use either iPTM or pDockQ as further support for the relevance of the proposed interaction. Such studies – which range from predicting protein quaternary structures to defining biological functions of peptides and intrinsically disordered regions (*Lim et al., 2023*; *Bret et al., 2024*; *Danneskiold-Samsøe et al., 2023*; *Marciano et al., 2022*; *Martin, 2024*; *Teufel et al., 2023*; *Pogozheva et al., 2023*) – highlight the value of using AF2 to nominate new candidate PPIs (i.e. in a screening setting), extending it beyond simply providing structures for fully validated protein pairings.

While the early adoption of iPTM or pDockQ has spurred their use in this screening, neither one was intended for this use case. Rather, both were developed with the goal of assessing model quality for AF2 structures, and the ability to distinguish between active and inactive structures on the basis of model quality was a convenient secondary benefit (*Evans et al., 2021*; *Bryant et al., 2022*). Due to this history, however, neither method was rigorously benchmarked in a manner that would set expectations for their performance in real-world applications.

To lay the groundwork for this study, we began by building a set of nonredundant active complexes and, from these, a set of highly compelling decoy complexes. The overarching philosophy behind the construction of this set was to realistically mimic conditions that would be encountered in future (prospective) screening applications. Notably, distinguishing actives from decoys in this set is markedly more challenging than in other (less realistic) benchmarks: e.g., pDockQ provided an AUC value of 0.87 in its first-reported classification experiment (*Bryant et al., 2022*), but only 0.71 when applied to this newer set (*Figure 3A*).

Developing this challenging and realistic set allowed us to train PPIscreenML, a tool expressly designed and optimized to be used in screening for PPIs. As expected, performance of PPIscreenML is superior to that of iPTM or pDockQ for this screening regime (*Figure 3A*). Moreover, PPIscreenML also makes use of subtle structural cues to provide its discriminative power, as evidenced from its impressive performance on the TNFSF selectivity benchmark (*Figure 4*).

Contemporaneously with our study, other groups have developed alternate methods for predicting pairs of interacting proteins. For example, the AF2Complex method also shows superiority over iPTM (*Gao et al., 2022*), albeit through testing on arbitrary collections of protein pairings (rather than on compellingly built decoys).

The natural extension of using AF2 and related methods to carry out structure-based screening for PPIs, naturally, is using these methods to define whole interactomes. Already, certain studies have applied these tools either within targeted protein families (*Weeratunga et al., 2024*; *Baryshev et al., 2023*) or to groups of proteins thought to belong together in large complexes (*Gao et al., 2022*; *Burke et al., 2023*; *Humphreys et al., 2021*). Importantly, careful consideration of the statistical measures needed for productively scaling assessments to whole proteomes must complement these studies; e.g., this will allow for defining thresholds that ideally balance precision and recall (*Table 1*), and also ensure that the expected results justify these larger-scale analyses.

There is also ample reason for optimism that the performance of PPIscreenML will further improve as the underlying AF2 models improve. Specifically, older versions of the AF2 software yielded slightly worse performance from PPIscreenML (*Figure 2—figure supplement 3*, *Figure 2—figure supplement 4*), but the difference in performance could be attributed to the fact that a (slightly) smaller proportion of the active complexes were correctly built. Thus, the true bottleneck in extending PPIscreenML may not lie with distinguishing correct versus incorrect interfaces. Rather, screening for interactions at whole-proteome scale may currently be instead limited by the fact that not all protein pairs are correctly modeled by AF2, and active pairings cannot be reliably identified if their structures are incorrectly built.

# Methods
## Model availability
All protein structures, code, and machine learning models described in this paper are available on GitHub (https://github.com/victoria-mischley/PPIScreenML copy archived at *Mischley, 2025*).

## Set of active complexes

Heterodimeric protein complexes were downloaded from the DockGround database (*Collins et al., 2022*). Complexes that were downloaded were published in the PDB (*Burley et al., 2023*) between up to 01/26/23 with a maximum of 3 Å resolution. No homodimers or antibody/antigen complexes were included in the dataset. The dataset was further filtered for a maximum of 30% sequence identity. Any heterodimeric complex with a total length greater than 1550 amino acids was filtered out of the dataset, due to our modeling capabilities at the time. Finally, the dataset was separated based on protein size. If a heterodimeric complex had at least one protein less than 50 amino acids, this complex was included in the protein-peptide dataset and excluded from the protein-protein dataset. After filtering, there were a total of 1481 protein-protein complexes and 456 protein-peptide complexes.

The sequences for these protein complexes were then downloaded directly from the PDB (*Burley et al., 2023*). Sequences were then used to model the complexes with ColabFold Batch (*Mirdita et al., 2022*) AF-Multimer v2 (*Evans et al., 2021*). Templates were used in building AF2 models, each model was recycled 12 times, and the AMBER relax step was included.

To avoid including inaccurate structural models in the set of actives, DockQ score (*Basu and Wallner, 2016*) was used to filter out incorrect models. DockQ (*Basu and Wallner, 2016*) is a continuous score that incorporates information from three scoring metrics, allowing for a holistic view of the complex's accuracy: the fraction of native interfaces, the ligand root mean square deviation, and iRMS. While DockQ is a continuous measure, a DockQ score of 0.23 or greater is considered at least 'acceptable' quality (*Basu and Wallner, 2016*). Therefore, we only included in the training set models with DockQ scores greater than 0.23. Out of the five models provided by AF for each heterodimeric complex, every model with a DockQ score greater than 0.23 was kept in the training set (*Figure 1—figure supplement 1*). Keeping all models with a DockQ score greater than 0.23 rather than taking only the model with the highest DockQ score allowed us to introduce some noise into our training dataset. However, our test set included all five AF models, as we would not know the correct multimeric structure a priori (*Figure 1—figure supplement 2*). Hence, utilizing all five models in the training set provides a more meaningful readout of expected performance in future prospective applications.

## Set of compelling decoy complexes

The set of active complexes was separated into their individual component proteins. For each protein, the TM-score (*Zhang and Skolnick, 2004*) was calculated between that protein and every other protein. In each complex, the two component proteins were replaced by their closest structural analogs: this provided a simple means to generate new complexes that resemble known complexes, but that are built using pairs of (presumably) noninteracting proteins. In cases for which pairs of structural analogs happened to come from the same alternate complex (i.e. the newly generated pairing corresponds to one of the active pairings), the analogs with the second-best TM-scores were used to generate a new decoy pairing.

After selection of the two proteins that would comprise a 'decoy' pairing, their sequences were used to build a set of AF2 models using the same settings as for the active complexes.

We did note that this method for building compelling decoys did give a slight bias toward larger proteins being used (*Figure 1—figure supplement 4*): this arises because larger decoy proteins are (slightly) more likely to have higher TM-score to the templates, and thus are slightly favored in building decoy complexes. Because this artifact could be leveraged by features that explicitly scale with the total size of the component proteins, such features were excluded from our study.

## Feature extraction

Structural features were extracted from AF2 models using three Python scripts (all available on GitHub): one for AF2-based features, one for Rosetta-based features (implemented using pyRosetta), and one for structural 'counting' features (implemented using Python package BioPandas). Residues were classified as interface residues if the Cβ atom (or Cα for glycine) was within 12 Å of a Cβ atom on the other chain.

To calculate iPAE top ¼, PAE values were calculated for all interacting residue pairs, then the highest ¼ of these values were averaged. To calculate tPAE top ¼, all PAE values for interchain residues-residue pairs were calculated, then the best (lowest) ¼ of these values were averaged.

## Data partitioning

Each protein pairing was randomly assigned to either training set/validation set/test set, with a 60/20/20 split. To avoid information leakage, all AF2 models for a given protein pairing were *together* placed in either training set/validation set/test set (rather than assigning individual AF2 models to different sets).

## Training and validation sets

The active complexes in the training set were filtered to include only AF2 models with greater than 0.23 DockQ score (relative to the corresponding crystal structure). Stratified fivefold cross-validation was used for the training and validation sets, such that all AF2 models for a given complex were together placed in either the training set or the validation set.

## Test set

Complexes used in the test set were separated from those in the training set and not used in any part of developing PPIscreenML.

Evaluation of PPIscreenML on the test set differed from the training/validation set in two ways (*Figure 1—figure supplement 2*), both intended to ensure that model performance on the test set will extend to future prospective applications. First, since experimentally derived structures will not be available, AF2 models in the test set were not filtered based on DockQ. Thus, the test set includes some active complexes for which the AF2 models are incorrectly built, as would occur in a prospective context. Second, rather than evaluating PPIscreenML scores for individual AF2 models, performance was evaluated by using PPIscreenML to classify each complex. Specifically, PPIscreenML was used to define scores for all five AF2 models, then the inferred class for the complex was assigned based on the maximum value from the five models.

## Machine learning classifier

Code for model training and optimization was adapted from a standard source (*Raschka et al., 2022*).

PPIscreenML was built using the XGBoost framework (*Chen and Guestrin, 2016*). Other machine learning frameworks evaluated included decision trees, random forest, K-nearest neighbors, logistic regression, AdaBoost, and Bagging (*Figure 2—figure supplement 1*). Sequential backward selection was then used to reduce the number of features from 57 to 7 (*Figure 2*). Finally, the model was hyperparameter tuned on the following parameters: maximum depth, number of estimators, learning rate, gamma, regularized lambda, and minimum child weight.

All model optimization (selecting the framework, feature reduction, and hyperparameter tuning) was carried out using the validation set to define the best model, and not ever the test set.

## Impact of AF2 version

Development of PPIscreenML and all performance benchmarking used models built with AF-Multimer v2.3. Performance is similar when PPIscreenML is applied to models built with AF-PTM (*Figure 2—figure supplement 3*) or with AF-Multimer v2.2 (*Figure 2—figure supplement 4*). In both cases, the AUC values were slightly worse than for AF-Multimer v2.3, but analysis of incorrect predictions shows that this difference can be traced to the fact that AF-PTM and AF-Multimer v2.2 build mis-docked models of active complexes slightly more often than AF-Multimer v2.3 does.

## TNFSF case study

All 18 TNFSF ligands were modeled against all 28 TNFSF receptors, for a total of 504 pairings (using sequences obtained from UniProt). Given the shared arrangement of subunits in all complexes described to date, a 3:3 stoichiometry was assumed in all cases. Because three receptor subunits separately engage the outside edges of the trimeric ligand, AF2 models included three copies of the ligand with one copy of the receptor.

Five models for each ligand/receptor pairing were built using the CF-batch AlphaFold-Multimer v2.3, with templates, 12 recycles, and AMBER relax step. Features from each structural model were extracted and used as input to PPIscreenML. Because PPIscreenML was trained on heterodimeric complexes, the three copies of the trimeric ligand were considered as the first entity and the receptor was considered as the second entity.

PPIscreenML results were compared with TNFSF binding specificity data drawn from a previously published comprehensive survey using flow cytometry (*Bossen et al., 2006*).

## Computational resources

Approximately 7700 GPU hours were used to build the AF2 models in this work. All models were built on ACCESS San Diego Supercomputer's Expanse NVIDIA V100 GPU processors (*Strande et al., 2021*).

## Acknowledgements

Parts of several figures were created using BioRender.com. This work was supported by the WM Keck Foundation, by the NIH National Institute of General Medical Sciences (R01GM141513), and by the National Institute of Biomedical Imaging and Bioengineering (through fellowship 1F30EB034594). This research was also funded in part through the NIH/NCI Cancer Center Support Grant P30CA006927. This work used the Extreme Science and Engineering Discovery Environment (XSEDE) allocation MCB130049, which is supported by National Science Foundation grant number 1548562. This work also used computational resources through allocation MCB130049 from the Advanced Cyberinfrastructure Coordination Ecosystem: Services & Support (ACCESS) program, which is supported by National Science Foundation grants 2138259, 2138286, 2138307, 2137603, and 2138296.

## Additional information

### Competing interests

Johannes Maier: Johannes Maier is affiliated with Triana Biomedicines, Lexington. The authors has no other competing interests to declare. Jesse Chen: Jesse Chen is affiliated with Triana Biomedicines, Lexington. The authors has no other competing interests to declare. The other authors declare that no competing interests exist.

### Funding

| Funder | Grant reference number | Author |
|---|---|---|
| W. M. Keck Foundation | | John Karanicolas |
| National Institutes of Health | R01GM141513 | John Karanicolas |
| National Institutes of Health | F30EB034594 | Victoria Mischley |
| National Institutes of Health | P30CA006927 | John Karanicolas |
| National Science Foundation | MCB130049 | John Karanicolas |

The funders had no role in study design, data collection and interpretation, or the decision to submit the work for publication.

### Author contributions

Victoria Mischley, Conceptualization, Data curation, Formal analysis, Validation, Investigation, Methodology, Writing – original draft, Writing – review and editing; Johannes Maier, Jesse Chen, Conceptualization, Methodology, Writing – review and editing; John Karanicolas, Conceptualization, Supervision, Funding acquisition, Validation, Methodology, Writing – original draft, Writing – review and editing

### Author ORCIDs

John Karanicolas https://orcid.org/0000-0003-0300-726X

Reviewer #1 (Public Review): https://doi.org/10.7554/eLife.98179.2.sa1
Reviewer #2 (Public Review): https://doi.org/10.7554/eLife.98179.2.sa2

Reviewer #3 (Public Review): https://doi.org/10.7554/eLife.98179.2.sa3

## Additional files

### Supplementary files
MDAR checklist

### Data availability
All protein structures, code, and machine learning models described in this paper are available on GitHub (https://github.com/victoria-mischley/PPIScreenML copy archived at *Mischley, 2025*).

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
