## [Editor Report · eLife assessment]

This study explores simple machine learning frameworks to distinguish between interacting and non-interacting protein pairs, offering **solid** computational results despite some concerns about dataset generation. The authors demonstrate a modest improvement in AlphaFold-multimers' ability to differentiate these pairs. Using a simple yet sound approach, this work is a **valuable** contribution to the challenging problem of reconstructing protein-protein interaction networks.

---

## [Referee Report · Reviewer #1 (Public Review)]

V.Mischley et al have applied several simple machine learning (ML)frameworks (which were widely used before the advent of deep learning methods) to distinguish (as the authors claimed) between interacting and non-interacting pairs. For this purpose, the authors have generated two sets of protein pairs, equal in their size (which is preferable for classification problems in ML). The first set comprises a non-redundant set of interacting proteins from the DOCKGROUND database, and the second set consists of presumably non-interacting protein pairs. Then, the authors trained and evaluated compared performance of the utilized ML frameworks using a set of well-described parameters. The authors also demonstrated the superior performance of their method in comparison to other metrics, such as ipTM and pdockQ. Finally, the authors applied their method to identify interacting pairs within the tumor necrosis factor superfamily. In general, the paper is well written, and the methodology applied is sound, however, I have a fundamental concern regarding the non-interacting set. As follows from the author's description, this set does not ensure that generated protein pairs do not interact as follows from the main paradigm of template-based docking (structurally similar proteins have similar binding modes). In my opinion, this set rather presents a set of non-cognate or weekly interacting protein pairs. That also explains the drop in performance for the pDockQ metric on the authors' set (AUC 0.71 in this paper opposite t0 0.87 in the original paper), as pDockQ was trained on the set of truly non-interacting proteins. In that respect, it would be interesting to see the performance of the authors' approach, but trained on the set described in the pDockQ paper (more or less the same set of interacting pairs but a different set of non-interacting proteins).

---

## [Referee Report · Reviewer #2 (Public Review)]

Summary:

In this paper, the authors train a simple machine learning to improve the ability of AlphaFold-multimers ability to separate interacting from non-interacting pairs. The improvement is small compared with the default AlphaFold score (AUROC from 0.84 to 0.88).

Strengths:

The dataset seems to be carefully constructed.

Weaknesses:

The comparison with the state of the art is limited.

- pDockQ comparison is (likely) incorrect (v2.1 should be used, not v1.0).

- Comparison with ipTM should be complemented with RankingConfidence (the default AF2-score).

- Several other scores than pDockQ have been developed for this task.

- Other methods (by Jianlin Chen) to "improve" quality assessment of AF2-models have been presented - these should at least be cited.

Lack of ablation studies:

- Quite likely the most significant contributor is the ipTM (and other scores from AF2). This should be analyzed and discussed.

Lack of data:

- The GitHub repository does not contain the models - so the data can not be examined carefully. Nor can the model be retrained.

- No license is provided for the code in the Git repository.

---

## [Referee Report · Reviewer #3 (Public Review)]

Due to AlphaFold's popularity, I see people taking the fact that AlphaFold predicted a decent protein complex structure between two proteins as strong support for protein-protein interaction (PPI) and even using such a hypothesis to guide their experimental studies. The scientific community needs to realize that just like the experimental methods to characterize PPIs, using AlphaFold to study PPIs has a considerate false positive and false negative rate.

Overall, I think it is solid work, but I have several concerns.

(1) In the benchmark set, the authors used about 1:1 ratio of positive (active) and negative controls. However, in real-life applications, the signal-to-noise ratio of PPI screening is very low. As they stated in their very nice introduction, there are expected to be "74,000 - 200,000" true PPIs in humans, whereas there are > 200,000,000 protein pairs. I am not suggesting that the authors need to make their tool able to handle such a high noise level, but at least some discussion along this line is helpful.

(2) The benchmark set from Dockground mostly consists of stable interactions that are actually relatively easily distinguished from non-interacting pairs. I suggest the authors test how well their tools will perform on weaker and transient interactions or discuss this limitation. For the more stable complexes, structural features at the interface are useful in predicting whether two proteins should interact, but I doubt this will be true for weaker and transient interactions.

(3) Given that the 1:1 benchmark set is a simplified task (see the first point) compared to real-life applications, the other task shown in this paper, i.e., the ligand/receptor pairings, seems to be more important. I think it is necessary to compare their tool against other simpler metrics for this more realistic task.